# Evaluation of community based surveillance in the Rohingya refugee camps in Cox's Bazar, Bangladesh, 2019

Elburg Van Boetzelaer[1]*, Samiur Chowdhury[1], Berhe Etsay[1], Abu Faruque[2], Annick Lenglet[3,4], Anna Kuehne[5,6], Isidro Carrion-Martin[5], Patrick Keating[5], Martins Dada[3], Jorieke Vyncke[5], Donald Sonne Kazungu[1], Maria Verdecchia[5]

1 Médecins Sans Frontières, Cox's Bazar, Dhaka, Bangladesh, 2 International Centre for Diarrhoeal Disease Research, Dhaka, Bangladesh, 3 Médecins Sans Frontières, Amsterdam, The Netherlands, 4 Department of Medical Microbiology, Radboudumc, Nijmegen, The Netherlands, 5 Médecins Sans Frontières, London, United Kingdom, 6 Médecins Sans Frontières, Berlin, Germany

* elburgvb@gmail.com

## Abstract

### Background

Following an influx of an estimated 742,000 Rohingya refugees in Bangladesh, Médecins sans Frontières (MSF) established an active indicator-based Community Based Surveillance (CBS) in 13 sub-camps in Cox's Bazar in August 2017. Its objective was to detect epidemic prone diseases early for rapid response. We describe the surveillance, alert and response in place from epidemiological week 20 (12 May 2019) until 44 (2 November 2019).

### Methods

Suspected cases were identified through passive health facility surveillance and active indicator-based CBS. CBS-teams conducted active case finding for suspected cases of acute watery diarrhea (AWD), acute jaundice syndrome (AJS), acute flaccid paralysis (AFP), dengue, diphtheria, measles and meningitis. We evaluate the following surveillance system attributes: usefulness, Positive Predictive Value (PPV), timeliness, simplicity, flexibility, acceptability, representativeness and stability.

### Results

Between epidemiological weeks 20 and 44, an average of 97,340 households were included in the CBS per surveillance cycle. Household coverage reached over 85%. Twenty-one RDT positive cholera cases and two clusters of AWD were identified by the CBS and health facility surveillance that triggered the response mechanism within 12 hours. The PPV of the CBS varied per disease between 41.7%-100%. The CBS required 354 full-time staff in 10 different roles. The CBS was sufficiently flexible to integrate dengue surveillance. The CBS was representative of the population in the catchment area due to its exhaustive character and high household coverage. All households consented to CBS participation, showing acceptability.

**Data Availability Statement:** All relevant data are within the paper and its Supporting Information files.

**Funding:** The authors received no specific funding for this work.

**Competing interests:** The authors have declared that no competing interests exist.

## Discussion

The CBS allowed for timely response but was resource intensive. Disease trends identified by the health facility surveillance and suspected diseases trends identified by CBS were similar, which might indicate limited additional value of the CBS in a dense and stable setting such as Cox's Bazar. Instead, a passive community-event-based surveillance mechanism combined with health facility-based surveillance could be more appropriate.

## Background

Close to one million Rohingya refugees have fled violence in Myanmar in successive waves of displacement since the early 1990s. Between August and October 2017, following a surge in violence in Myanmar's Rakhine state, there was an acute exodus of 742,000 Rohingya refugees across the border to Bangladesh [1]. As of November 2019, 914,998 Rohingya refugees are residing in refugee camps in the Cox's Bazar district [2]. Since 1990, Médecins sans Frontières (MSF) worked in the earlier established refugee camp and they scaled up their activities in August 2017 to respond to the influx of refugees. MSF established a secondary healthcare facility, a specialized Diphtheria clinic and community outreach activities including health education, Community Based Surveillance (CBS), traditional birth attendants (TBA) and women's health promotors.

Indicator-based CBS is often established in addition to health facility based surveillance during population displacement events. It may facilitate early detection of epidemic prone diseases in hard-to-reach communities or areas where the availability and accessibility of health centers is lacking [3]. The design of CBS varies according to context and surveillance objectives. Some CBS train community volunteers to actively search for cases using pre-defined case definitions (thus indicator based data collection), while others use an event-based approach. The latter will be less specific but potentially more useful in detecting a wider variety of public health events, reaching remote areas and guiding outbreak response [4].

In order to monitor the refugee health status and to respond in a timely manner to emerging public health threats, MSF established surveillance activities in Cox's Bazar in August 2017 with the aims to: 1. detect and timely respond to suspect cases of epidemic prone diseases at health facilities or in the population; 2. monitor community based mortality (including still births and neonatal deaths); 3. monitor community level water and sanitation indicators; 4. identify pregnant women to allow for targeted follow-up by TBAs; 5. monitor population movement. The surveillance system included two components: health facility based reporting (from two secondary health structures) and indicator-based CBS. In addition, an early warning and response mechanism was established through which suspected cases of diseases under surveillance that were detected by MSF surveillance systems were investigated by MSF led response teams and if verified and validated reported to the WHO-led Early Warning Alert and Response System (EWARS). We describe and evaluate the indicator-based CBS and early warning and response mechanism that MSF implemented in Cox's Bazar. Since its establishment in August 2017, the CBS system underwent some adaptations, including the integration of an alert and response component that was fully functional in April 2019. We therefore have focused on the time period from epidemiological week 20 (12 May 2019) until 44 (2 November 2019).

## Methods

The concept note for this evaluation was submitted to the Research Committee of Medecins sans Frontieres prior to the evaluation and drafting of the manuscript. As it is a retrospective

description of an intervention, which only includes aggregated health data from routine reporting, this publication was exempted from ERB review. As the evaluation was based on anonymized, aggregated data, consent was not obtained specifically for the evaluation. However, all households included in the surveillance system gave oral consent to be included into the surveillance system.

## Indicator-based community based surveillance

**Surveillance population and diseases under surveillance.** The CBS included 13 subcamps with a total surface of 12 square kilometers (Fig 1).

**Diseases and information under surveillance.** The CBS system used standard case definitions (Table 1). In addition to data on suspected cases of disease, surveillance workers also collected other data to guide public health interventions (Fig 2). The surveillance worker enquired whether there were any household members with signs and symptoms that met the case definition of diseases under surveillance on the day of their visit. The surveillance worker then referred all suspected cases of disease to the nearest health facility and recorded the household number to inform further case investigations by the Epi Alert Team or Medical Response Team (see "Alert and Response" section). All households (100%) consented to be included in MSF's CBS.

**Data collection.** Prior to CBS implementation, households in the catchment area were assigned unique household identification numbers that were written on the doorposts. The CBS was exhaustive as each household in the catchment area was visited by a surveillance worker once during each surveillance cycle of four weeks. During the first two weeks of the surveillance cycle, the surveillance worker visited evenly numbered households and odd

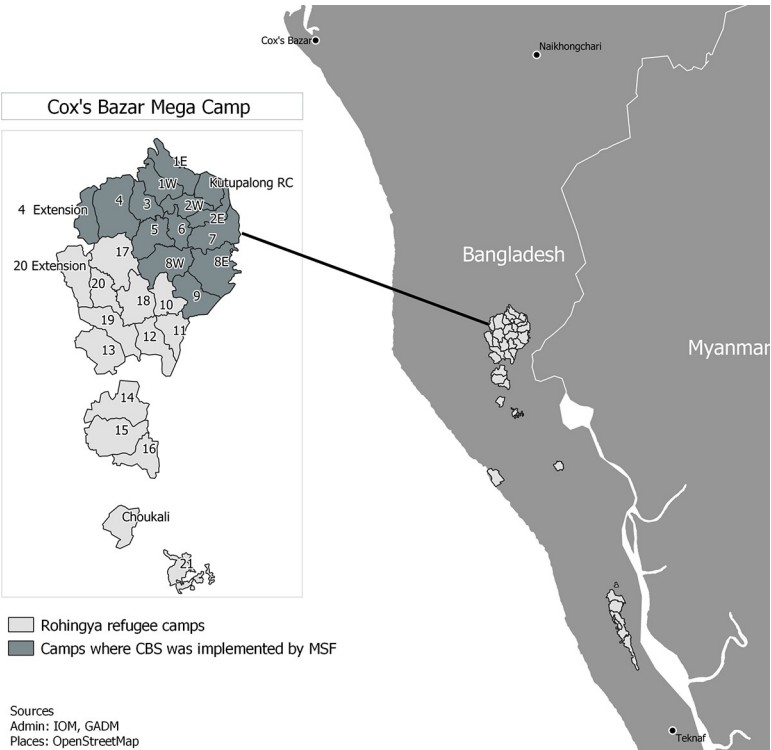

**Fig 1. Map of catchment area of MSF's CBS system in Cox's Bazar, Bangladesh, week 20–44, 2019 (source: MSF).**

**Table 1. CBS case definitions syndromes/diseases under surveillance by MSF in Cox's Bazar, Bangladesh, week 20–44, 2019 [5].**

| | Case definitions used by Community Based Surveillance workers | Case definitions used at health facility | Alert threshold |
|---|---|---|---|
| Acute Jaundice Syndrome (AJS) | Acute onset yellow discoloration of the skin and/or whites of the eyes, dark urine, anorexia, malaise, extreme fatigue and abdominal tenderness | Acute onset yellow discoloration of the skin and/or sclera of the eyes, dark urine, anorexia, malaise, extreme fatigue and abdominal tenderness | One probable case of Hepatitis E (RDT+) |
| | | Probable Hepatitis E: AJS + positive RDT for Hepatitis E | |
| Acute Flaccid Paralysis (AFP) | Child under 15 years of age with acute onset of focal weakness or paralysis OR Person with paralytic illness at any age when polio suspected, characterized by flaccid (reduced tone), without other obvious causes (e.g. trauma) | Child under 15 years of age with acute onset of focal weakness or paralysis OR Person with paralytic illness at any age when polio suspected, characterized by flaccid (reduced tone), without other obvious causes (e.g. trauma) | A single case of paralysis in a child under 15 years of age |
| Acute Watery Diarrhoea (AWD) | Any patient presenting with 3 or more liquid stools and/or vomiting for the last 24 hours | Any patient presenting with 3 or more liquid stools and/or vomiting for the last 24 hours | One probable case of cholera (RDT+) OR A sudden increase in adults with acute watery diarrhoea and/or an inversion in the proportion of diarrhoeal cases between adults and children |
| | | Probable Cholera: AWD + positive RDT for Cholera | |
| Dengue | Fever AND Two of the following: • Nausea, vomiting • Rash • Aches and pains | Dengue without warning signs Live in/travel to dengue endemic area, AND Fever AND Two of the following: • Nausea, vomiting • Rash • Aches and pains • Leukopenia • Positive tourniquet test | One probable case of dengue/severe dengue (RDT+) |
| | Dengue with warning signs Dengue, as defined above, with any of the following: • Abdominal pain or tenderness • Persistent vomiting | Dengue with warning signs Dengue, as defined above, with any of the following: • Abdominal pain or tenderness • Persistent vomiting • Clinical fluid accumulation (ascites, pleural effusion) • Mucosal bleeding • Lethargy, restlessness • Liver enlargement >2cm • Laboratory: increase in HCT concurrent with rapid decrease in platelet count | |
| Diphtheria | Sore throat and hoarseness of voice AND Swollen glands (bull neck) AND Difficulty breathing or rapid breathing AND Nasal discharge AND Low grade fever AND Malaise AND A thick, grey membrane covering throat and tonsils | Sore throat and hoarseness of voice AND Swollen glands (enlarged lymph nodes) AND Difficulty breathing or rapid breathing AND Nasal discharge AND Low grade fever AND Malaise AND A thick, grey membrane covering throat and tonsils | One suspected case of Diphtheria |
| Measles | Fever AND Generalised rash AND One of the following signs: cough or runny nose or red eyes | Fever AND Generalised rash (non-vesicular) AND One of the following signs: cough or coryza or conjunctivitis | 5 suspected cases reported by a single geographic unit in a one month period |
| Meningitis | Children under one year of age with: Fever AND Bulging fontanelle OR Rash | Children under one year of age with: Fever AND Bulging fontanelle OR petechial rash | 3 suspected cases/100,000 inhabitants /week (minimum of 2 cases in one week) |
| | Children over one year of age and adults with: Sudden onset of fever AND Neck stiffness OR Rash | Children over one year of age and adults with: Sudden onset of fever AND Neck stiffness and/or other meningism OR Rash | |

numbered households in the second two weeks. A census was conducted by MSF in March 2019 with population denominator estimates assigned to each camp. For subsequent surveillance cycles, population denominators were adjusted based on reported births, deaths and relocations (as collected through CBS).

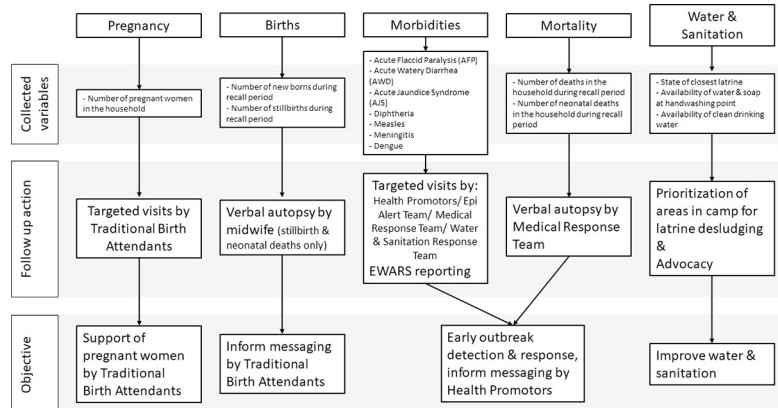

**Fig 2. Use of CBS data to guide targeted programmatic interventions by MSF in Cox's Bazar, Bangladesh, week 20–44, 2019.**

Surveillance workers and team leaders, members of the Epi Alert Team and of the Water and Sanitation Response Team were recruited from the Rohingya camp population, the main recruitment criterion was literacy. Surveillance workers had been working with MSF since September 2017 and received refresher trainings every six months. The content of the refresher trainings included case definitions, correct use of data collection forms and referral pathways. The Epi Alert Team participated in a three-day initial training and quarterly refresher trainings. The Medical Response Team and Outreach Midwifes and one epidemiologist were Bangladeshi nationals and there was one epidemiologist of other nationality involved. As the Medical Response Team were formally trained medical assistants, they only received a half-day training on case definitions, correct use of data collection forms and referral pathways.

Surveillance workers used paper forms to record data. Team leaders filled out daily summary forms on which they recorded the household numbers where surveillance workers identified suspected cases of disease or deaths. Due to the large number of surveillance workers (140 persons in total) and the material and logistical resources that would require, data collection was paper based rather than electronic.

**Alert and response.** Daily summary forms that were completed by surveillance team leaders informed follow-up visits by the Epi Alert Team, Medical Response Team and Outreach Midwifes. The Epi Alert Team—a 13-person team tasked with the follow-up within 24 hours of suspected cases of AWD, AJS, Diphtheria and Measles, see Table 2 - and the Medical Response Team—a 4-person clinically trained team tasked with the follow-up of suspected cases of AFP and Meningitis, see Table 2 - conducted a household visit for each suspected case during which they did a case investigation to confirm that the suspected case met the case definition. The suspected case was then referred to the nearest health facility for testing and reported into the WHO-led EWARS system at the health facility level [6]. The nearest health facility was often not an MSF operated health facility, which made effective referral tracking impossible. If any deaths were reported, surveillance workers recorded the household number to inform verbal autopsies by the Medical Response Team or the Outreach Midwifes. Mobile devices with KoboCollect (KoboToolbox, kobotoolbox.com) were used to record case investigations and verbal autopsies to allow for timely response if necessary. The household numbers where newly identified pregnant women resided were communicated to TBAs who then visited the households the next day to provide council to the women on antenatal care.

**Data entry and data analysis.** Relocation, mortality and birth data were entered in a Microsoft Excel database by two full time data encoders, and analyzed using STATA (V13,

**Table 2. Description of human resources engaged in MSF's public health surveillance in Cox's Bazar, Bangladesh, week 20–44, 2019.**

| Position | Description of tasks | Number of staff |
|---|---|---|
| Surveillance Worker | Monthly household visits to detect suspected cases of disease; relocations; mortality (incl. stillbirths and neonatal deaths) | 140 |
| Outreach Team Leader | Daily consolidation of data collected by surveillance workers and disseminate for follow-up, supervision of 4 surveillance workers per Outreach Team Leader | 35 |
| TBAs | Follow up on households with pregnant women (identified by surveillance workers) | 70 |
| Health Promoters (HP) | Follow up on households with suspected cases of waterborne diseases (identified by surveillance workers) | 70 |
| Epi Alert Team | Case investigation of suspected cases of diseases (AJS, AWD, diphtheria, measles) (identified by surveillance workers); active case finding in case of alert | 13 |
| Medical Response Team | Case investigation of suspected cases of diseases (AFP, meningitis) (identified by surveillance workers), general verbal autopsies | 6 |
| Outreach Midwifes | Stillbirth, neonatal and maternal death verbal autopsies | 3 |
| WatSan Response Team | In case of alert of cluster of waterborne disease or probable case of cholera or hepatitis E: bucket chlorination, latrine cleaning, soap distribution, hygiene promotion sessions | 12 + 1 supervisor |
| Data encoders | Weekly entry of surveillance data | 2 |
| Epidemiologists | Management of public health surveillance system, Data cleaning, analysis and reporting | 2 |

Stata corp). Mortality and births rates and surveillance coverage were included in an MSF internal monthly epidemiological bulletin. Data on suspected cases of diseases, verbal autopsies and water and sanitation indicators were entered into KOBOCollect by the Epi Alert Team and Medical Response Team while they were conducting home visits, and included in a biweekly MSF internal epidemiological bulletin. An example of a data collection form for suspected AWD that was filled out by the Epi Alert Team can be found in S1 File. All data was entered once. Epidemiologists checked the data from the Epi Alert Team and Medical Response Team daily to identify any clusters of suspected cases that warranted an immediate alert response.

## Passive health facility surveillance

**Population and diseases under surveillance.** At two MSF health facilities, clinicians collected data during consultations on suspected and probable—i.e. positive rapid diagnostic tests (RDTs)—cases of diseases under surveillance. The catchment population of the CBS and health facility surveillance differed as the catchment area of the health facilities was larger than that of the CBS as they also provide services to the host community. One of the health facilities was a referral hospital while the other was a specialized clinic providing Diphtheria treatment for the whole megacamp and host community. The same diseases that were included in the CBS were under surveillance at the health facilities using the same MSF case definitions (Table 1).

**Data collection, analysis & reporting.** Line lists were paper-based and entered once per week by a data encoder in a Microsoft Excel database for MSF monitoring purposes, as well as in the EWARS system [6]. At both health facilities, RDTs were used to identify probable cases of Cholera and Hepatitis E. If a probable case was identified, the clinician on duty immediately

contacted the epidemiologists by phone who in turn would activate the alert response mechanism (see "Alert and Response" section).

Data were analyzed in Microsoft Excel and disseminated through monthly internal MSF epidemiological bulletins.

**Alert and response.** The response mechanism for alerts that came from the MSF health facility surveillance consisted of several steps. Firstly, the epidemiologists reported probable cases of Cholera, Hepatitis E, AJS as well as clusters of AJS and AWD into the EWARS mechanism. This activated the WHO-led Joint Assessment Team (JAT) who conducted a case investigation. The JAT consisted of representatives of the health sector, of the water, sanitation and hygiene (WASH) sector and of the International Centre for Diarrhoeal Disease Research, Bangladesh (icddr,b). The samples were sent to the reference lab in Dhaka for pathogen screening. Secondly, MSF's Epi Alert Team was deployed to the affected household and surrounding area to conduct active case finding and contact tracing in a radius of 20–30 houses around affected households. The Epi Alert Team referred additional suspected AWD and Hepatitis E cases to the nearest health facility for testing and treatment. For each additional suspected case, the Epi Alert Team filled out a case investigation form on a mobile device, using KoboCollect (S1 File). Based on active case finding, the icddr,b acquired stool samples of additional community based suspected cases of AWD or Cholera. Simultaneously, MSF's Water and Sanitation Response Team was deployed to the affected household and surrounding area to conduct bucket chlorination, latrine cleaning, soap distribution and hygiene promotion sessions for seven days.

An overview of the public health surveillance system, alert and response system as implemented by MSF can be found in Fig 3.

## Public health surveillance system: Evaluation

For the CBS and the alert and response mechanism we evaluated the following attributes of the public health surveillance system: usefulness, simplicity, flexibility, acceptability, Positive Predictive Value (PPV), representativeness, timeliness and stability (Definitions and measurement of surveillance system attributes considered in this evaluation can be found in Table 3). The period under evaluation was epidemiological weeks 20 to 44. We used health facility surveillance data as the relevant gold-standard of surveillance data with which to compare trends of the CBS data.

# Results

On average 97,340 households were included per CBS surveillance cycle (range: 88,184–100,535), consisting of on average 548,739 persons (range 545,785–553,249) (Table 4). Each surveillance worker covered on average 36 households per day (range 32–40) and 714 households per month (range 628–806).

## Positive predictive value

The PPV of a notified suspected case to meet the verified case definition according to the Epi Alert Team or Medical Response Team varied per disease. The highest PPV was calculated for AFP (100%; 28/28) followed by AWD (88.8%; 2,528/2,848), and measles (73.7%; 101/137) and the lowest PPV was calculated for Meningitis (50%; 1/2) and Diphtheria (41.7%; 177/425) (Table 5).

Available health facility surveillance data from two MSF health facilities and data from the CBS (suspected cases identified by the surveillance workers and verified by the Epi Alert Team) are shown for AWD, AJS, Diphtheria and measles (Figs 4–7).

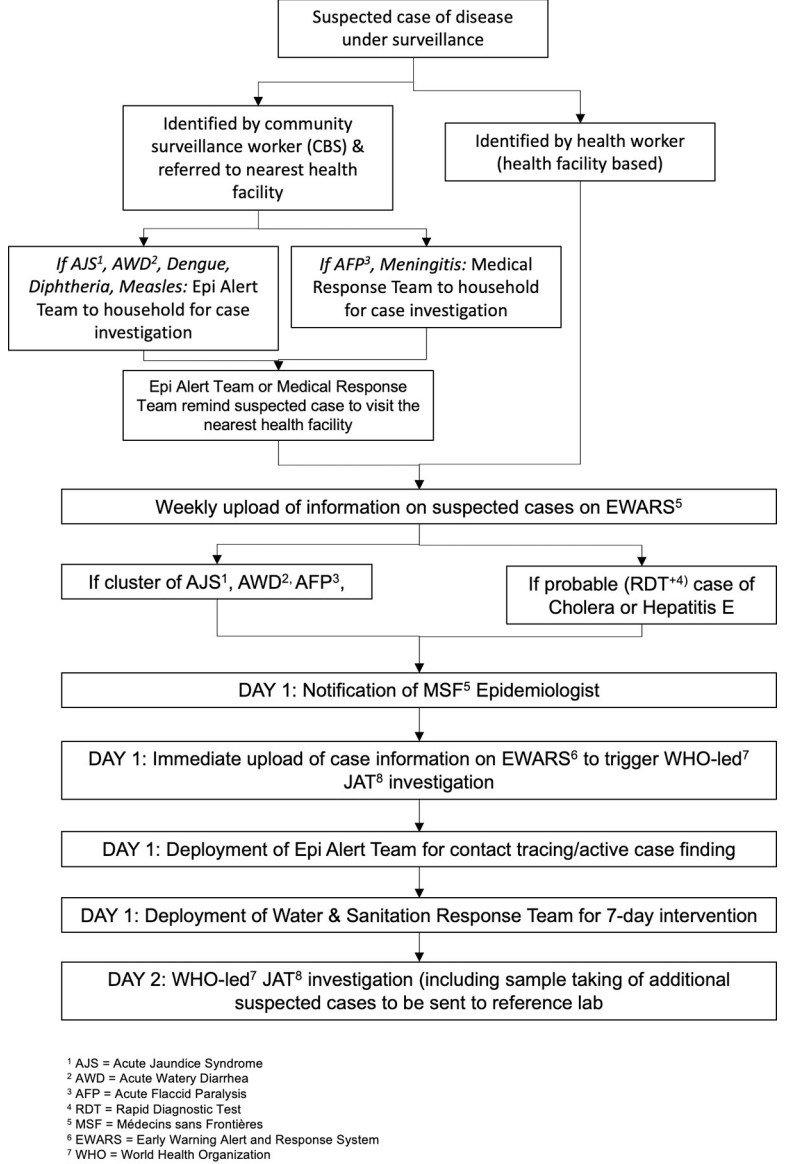

[1] AJS = Acute Jaundice Syndrome
[2] AWD = Acute Watery Diarrhea
[3] AFP = Acute Flaccid Paralysis
[4] RDT = Rapid Diagnostic Test
[5] MSF = Médecins sans Frontières
[6] EWARS = Early Warning Alert and Response System
[7] WHO = World Health Organization
[8] JAT = Joint Assessment Team

**Fig 3. Overview of MSF's public health surveillance, alert and response system in Cox's Bazar, Bangladesh, week 20–44, 2019.**

## Usefulness

Twenty-one probable (RDT positive) cholera cases were identified by health facility based surveillance and triggered the alert response mechanism. Per probable cholera case, MSF's Epi Alert Team visited on average 335 surrounding households (range 202–520). For each alert, MSF's Water and Sanitation Response Team conducted bucket chlorination (6–10 water points), cleaning of the latrines (15–22 latrines), soap distribution (100–120 pieces) and hygiene promotion sessions (250–300 households) in the affected area. The reporting of the probable cholera cases into EWARS triggered a case investigation by the WHO-led JAT. Of the 21 RDT positive Cholera cases, 17 were lab confirmed with culture by icddr,b. All the additional stool samples that were taken from additional suspected cases came back negative for Cholera culture.

**Table 3. Definitions and measurement of surveillance system attributes [7].**

| Attribute | Definition | Measurement | Data Source |
|---|---|---|---|
| Predictive Positive Value (PPV) | Positive Predictive Value (PPV) is the proportion of reported cases that actually have the health-related event under surveillance | • Proportion of suspected cases identified by surveillance workers and validated as meeting the case definition by the Epi Alert Team or Medical Response Team | • Data collected by surveillance workers on suspected cases<br><br>• Data collected by Epi Alert Team and Medical Response Team during verification |
| Usefulness | Ability of the public health surveillance system to detect outbreaks and trigger a response | • Proportion of probable cases detected that triggered a response<br><br>• Proportion of clusters of suspected cases detected that triggered a response | • Alert & Response logbook |
| Timeliness | Timeliness reflects the speed between steps in a public health surveillance system | • Time between identification of suspected case by surveillance worker and response<br><br>• Time between identification of probable case by health facility and response | • Alert & Response logbook |
| Simplicity | The simplicity of a public health surveillance system refers to both its structure and ease of operation. | Amount and type of data necessary to establish that the health-related event has occurred (i.e., the case definition has been met);<br><br>• Level of integration with other systems;<br><br>• Method of collecting the data, including number and types of reporting sources, and time spent on collecting data;<br><br>• Amount of follow-up that is necessary to update data on the case;<br><br>• Method of managing the data, including time spent on transferring, entering, editing, storing, and backing up data;<br><br>• Methods for analysing and disseminating the data, including time spent on preparing the data for dissemination;<br><br>• Staff training requirements; and<br><br>• Time spent on maintaining the system. | • Data collection forms (filled out by surveillance workers, Epi Alert Team and Medical Response Team) for CBS and health facility surveillance<br><br>• Data management tool (Microsoft Excel database)<br><br>• Biweekly epidemiological bulletins<br><br>• Human Resources data and structure<br><br>• Training agendas, materials and attendance sheets |
| Flexibility | A flexible public health surveillance system can adapt to changing information needs or operating conditions with little additional time, personnel, or allocated funds. Flexible systems can accommodate, for example, new health-related events, changes in case definitions or technology, and variations in funding or reporting sources. | • How the system has responded to a new demand | • Data collection forms (filled out by surveillance workers, Epi Alert Team and Medical Response Team) for CBS<br><br>• Data management tool (Microsoft Excel database)<br><br>• Biweekly epidemiological bulletins<br><br>• Training • agendas, materials and attendance sheets |
| Representativeness | A public health surveillance system that is representative accurately describes the occurrence of a health-related event over time and its distribution in the population by place and person | • Comparing the characteristics of reported events to all such actual events, including characteristics of the population, including, age, socioeconomic status, access to health care, and geographic location | • Data collection forms (filled out by surveillance workers, Epi Alert Team and Medical Response Team) for CBS and health facility surveillance |
| Acceptability | The willingness of persons and organizations to participate in the surveillance system | • Subject or agency participation rate (if it is high, how quickly it was achieved);<br><br>• Physician, laboratory, or hospital/facility reporting rate; and | • Data collection forms (filled out by surveillance workers, Epi Alert Team and Medical Response Team) for CBS and health facility surveillance |
| Stability | Stability refers to the reliability (i.e., the ability to collect, manage, and provide data properly without failure) and availability (the ability to be operational when it is needed) of the public health surveillance system | • the number of unscheduled outages and down times for the system's computer;<br><br>• the costs involved with any repair of the system's computer, including parts, service, and amount of time required for the repair;<br><br>• the percentage of time the system is operating fully;<br><br>• the desired and actual amount of time required for the system to collect or receive data;<br><br>• the desired and actual amount of time required for the system to manage the data, including transfer, entry, editing, storage, and back-up of data; and<br><br>the desired and actual amount of time required for the system to release data. | • Data collection forms (filled out by surveillance workers, Epi Alert Team and Medical Response Team) for CBS<br><br>• Data management tool (Microsoft Excel database)<br><br>• Biweekly epidemiological bulletins |

**Table 4. Catchment population MSF's CBS in Cox's Bazar, Bangladesh, week 20–44, 2019.**

| Surveillance cycle | Epi weeks | # Households covered/visited during surveillance cycle | # Households included in CBS catchment area | Est. total population | Est. population at beginning of surveillance cycle | | Coverage (percentage of included households visited by surveillance workers) |
|---|---|---|---|---|---|---|---|
| | | | | | < 5 years | > = 5 years | |
| 1 | 20–23 | 95,167 | 111,688 | 545,785 | 116,072 | 429,713 | 85.2 |
| 2 | 24–27 | 100,535 | 104,289 | 547,205 | 116,809 | 430,396 | 96.4 |
| 3 | 28–31 | 99,317 | 104,434 | 548,625 | 117,546 | 431,079 | 95.1 |
| 4 | 32–35 | 99,900 | 115,225 | 550,191 | 118,357 | 431,834 | 86.7 |
| 5 | 36–39 | 98,653 | 101,183 | 551,755 | 119,163 | 432,592 | 97.5 |
| 6 | 40–44 | 98,668 | 103,970 | 553,249 | 120,280 | 432,969 | 94.9 |

During the reporting period, two additional alert responses were activated due to clusters of suspected AWD that were identified by the CBS. For both alert responses MSF's Water and Sanitation Response Team was deployed. Both clusters were also reported to the WHO, who initiated a JAT investigation and advocated with other WASH actors to improve their interventions.

Community based mortality surveillance showed rates that were below emergency thresholds (CMR 0.05–0.07/10,000 persons/day; U5MR 0.04–0.14/10,000 persons/day) and informed verbal autopsies by the Medical Response Team of Midwifes (for stillbirths, neonatal and maternal deaths), the data collected through the verbal autopsies informed MSF's programming such as targeted health promotion messaging for pregnant women regarding the risks of over-utilization of Oxytocin before and during labor.

## Timeliness

When comparing the timeliness of the different elements of the health facility based surveillance and CBS, the main difference is the delay between the identification of a suspected case and the notification of the epidemiologist and Epi Alert Team or Medical Response Team

**Table 5. Positive predictive value of a notified suspected case to meet the case definition in MSF's CBS in Cox's Bazar, Bangladesh, week 20–44, 2019.**

| Disease | Number of suspected cases identified by CBS | Number of cases that resulted in any public health action (referral for case management OR active case finding/contact tracing OR health promotion OR water & sanitation intervention, or any combination) | Number of suspected cases verified by Epi Alert Team or Medical Response Team as 'suspected' | Percentage of identified suspected cases by CBS and verified as 'suspected' by Epi Alert Team or Medical Response Team |
|---|---|---|---|---|
| Acute diarrhoea (watery) | 2,848 | 2,848 | 2,528 | 88.8 |
| Acute Jaundice Syndrome | 684 | 684 | 364 | 53.2 |
| Acute Flaccid Paralysis | 28 | 28 | 28 | 100 |
| Dengue | 30 | 30 | 21 | 70 |
| Diphtheria (suspected) | 425 | 425 | 177 | 41.7 |
| Measles (suspected) | 137 | 137 | 101 | 73.7 |
| Meningitis (suspected) | 2 | 2 | 1 | 50 |

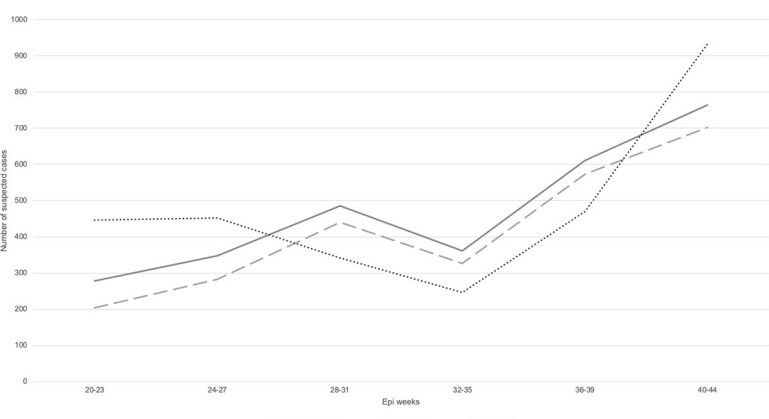

**Fig 4. Trends in suspected AWD cases compared between two MSF health facilities, surveillance workers and Epi Alert Team in Cox's Bazar, Bangladesh, week 20–44, 2019.**

through health facility based surveillance versus through CBS (Table 6). Notification of teams of suspected cases identified through health facility based surveillance occurred within 2 hours via phone (which was possible due to the relatively small number of suspected or probable cases identified at the health facilities). Epidemiologist and the Epi Alert Team or Medical Response Team were notified of suspected cases identified through the CBS at the end of each working day after collating the daily data. After the notification of the epidemiologist and Epi Alert Team or Medical Response Teams, the timelines were the same between the health facility based surveillance and CBS.

## Simplicity

While this surveillance system generated data that informed programming and enabled early detection of epidemic prone disease, the surveillance system was complex and required 354 staff in 10 different roles (Table 2). There were 140 surveillance workers, of which one female surveillance worker. All surveillance workers were literate. There were team leaders and supervisors in place to provide regular supervision visits for the surveillance workers, and the

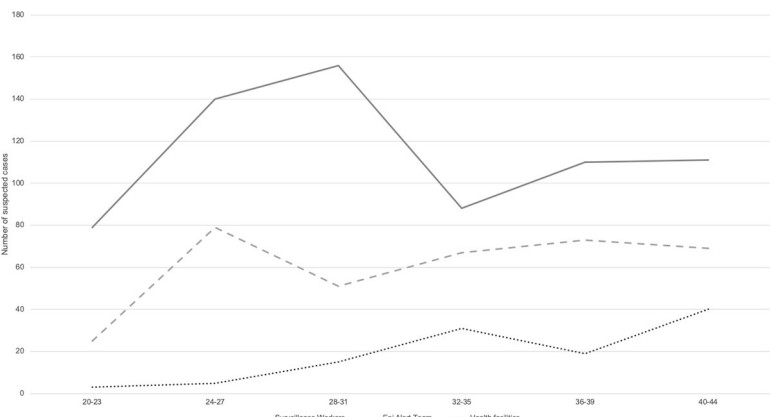

**Fig 5. Trends in suspected AJS cases compared between two MSF health facilities, surveillance workers and Epi Alert Team in Cox's Bazar, Bangladesh, week 20–44, 2019.**

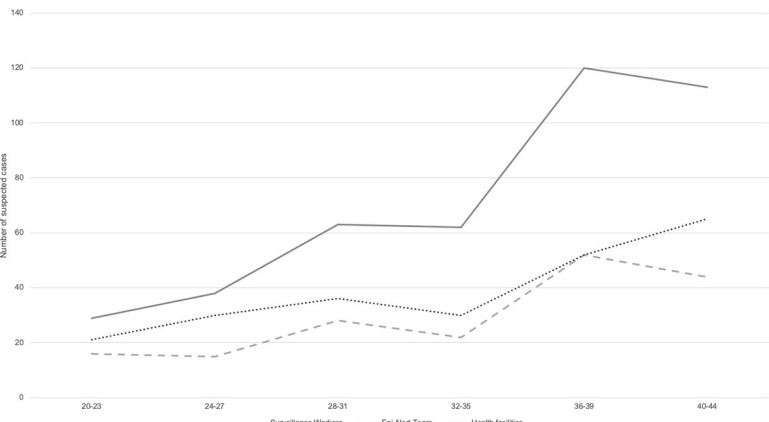

**Fig 6. Trends in suspected diphtheria cases compared between two MSF health facilities, surveillance workers and Epi Alert Team in Cox's Bazar, Bangladesh, week 20–44, 2019.**

epidemiologists conducted supervision of the Epi Alert Team and the Medical Response Team. Each team leader supervised four surveillance workers.

The integration with other systems, including WHO's EWARS was easy to manage by appointing one focal point amongst the data encoders to submit reports and establishing good coordination among WHO and MSF epidemiologists.

## Flexibility

When there was an increase of dengue cases identified in Cox's Bazar in September 2019, dengue was added to the list of diseases under surveillance (Table 1). This was incorporated following a 0.5-day training of surveillance workers on the dengue case definition. In addition, the CBS was designed with a built-in flexibility to allow for periodic rotations of water and sanitation indicators.

## Representativeness

The CBS was exhaustive, as all households in the catchment area were included. Surveillance coverage was lowest in epi weeks 20–23 (85.2%) and 32–35 (86.7%), as two major Islamic

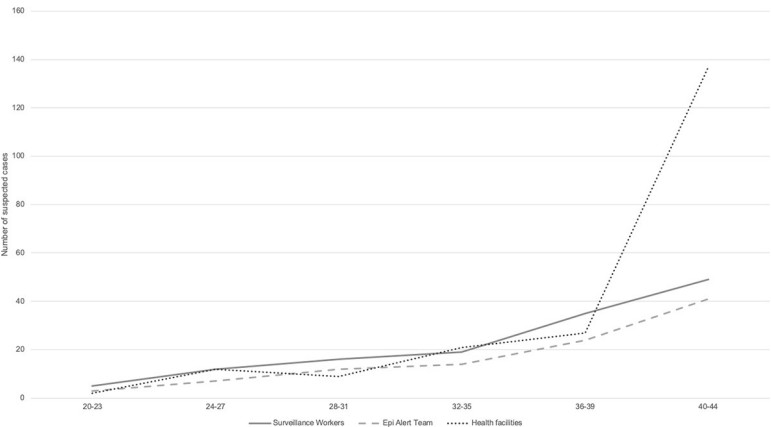

**Fig 7. Trends in suspected measles cases compared between two MSF health facilities, surveillance workers and Epi Alert Team in Cox's Bazar, Bangladesh, week 20–44, 2019.**

**Table 6. Timeliness of MSF's health facility based surveillance versus CBS in Cox's Bazar, Bangladesh, week 20–44, 2019**[*]**.**

| Average time between: | Health facility based surveillance (21 RDT + Cholera cases) | Community Based Surveillance (2 AWD clusters) |
|---|---|---|
| Identification of suspected case—Notification of epidemiologist | 1 hour | Within 8 hours (At end of working day) |
| Identification of suspected case—Notification of Epi Alert Team or Medical Response Team | 2 hours | Within 8 hours (At end of working day) |
| Notification of Epi Alert Team—MSF response *(case investigation, active case finding, water & sanitation response intervention, targeted health promotion messaging)* | 1–12 hours (depending on time of notification) | 1–12 hours (depending on time of notification) |
| Notification of epidemiologist—EWAR reporting | 1 hour | 1 hour |
| EWAR reporting—WHO's JAT investigation | 1–5 days | 1–5 days |

[*]Delay of onset of symptoms and health facility visit unaccounted for in this table.

public holidays coincided in the respective periods. The other surveillance cycles reached over 90% coverage, with the highest coverage (97.5%) in epi weeks 36–39 (Table 4).

## Acceptability

All households in the catchment area consented to be included in the CBS and to monthly household visits.

## Stability

No interruptions were reported during the implementation of the surveillance system.

## Discussion

We described the indicator-based CBS and early warning and response mechanism that MSF implemented in Cox's Bazar. This evaluation shows that the CBS system and the alert and response mechanism allowed for timely response to cases of epidemic prone diseases but was resource intensive. This is in line with findings from other contexts that CBS requires high numbers of human resources, ongoing monitoring and validation to maintain sensitivity of the system and regular population counts to ensure the accuracy of denominator estimates [8, 9]. The objective of adding indicator based CBS to the existing health facility based surveillance was not only to enable early detection of epidemic prone diseases at the community level, but also to provide targeted response and follow-up of suspected cases of disease to enable early access to treatment and therewith decrease morbidity and mortality.

As has been seen in other contexts, CBS implemented by community volunteers without formal medical training may increase the generation of false alerts [10] and decreasing the PPV (percentage of suspected cases that met the case definition). The Epi Alert Team and Medical Response Team were established to validate the suspected cases of disease that were identified by surveillance workers. The PPV of CBS varied per disease. This could be explained by the complexity of differentiating between diseases when signs and symptoms are non-specific (e.g. meningitis) or easily confused by someone who has not received formal medical training (measles and varicella). It is expected that training improved the ability of community volunteers to detect true disease, as has been noted in other contexts [3, 11]. However, even for

trained physicians diagnoses without laboratory confirmation can remain challenging (*Polonsky et al*, *in press*).

We were unable to assess the sensitivity of the system as we did not have a gold standard in which health facility staff verified whether the suspected cases that were identified by the surveillance workers met the case definition, nor did we know the real occurrence of cases in the catchment area. While the trends of reported suspected cases from all the components of the surveillance system were similar, and could be an indicator of data quality and validity, it must be noted that the catchment population of the CBS and health facility surveillance differed as the health facilities were also visited by patients who did not live in the CBS catchment area. Surveillance workers only visited each household once per month and only reported suspected cases that were experiencing signs and symptoms during the household visit (not retrospectively during the past month). During the four weeks in between surveillance visits, the household had to self-refer to a health facility nearby if a household member fell ill. Considering the apprehension toward health facilities from the Rohingya population [12], suspected cases may have gone undetected as people may have preferred self-medication as opposed to seeking care at a health facility.

We were only able to assess the time between the notification of the suspected case, verification and response. We were unable to assess the time between symptom onset and the identification of a suspected case by the surveillance workers. Even so, the timeliness of the MSF reporting and intervention was acceptable.

As it has been mentioned in other contexts regarding the usefulness of a CBS system, monthly household visits by surveillance workers contributed to a higher visibility of MSF in the community. This provided the opportunity for surveillance workers to identify cases early and to provide referrals to health facilities or other services with the aim to reduce morbidity and mortality. In addition, it allowed MSF to get a better understanding of population feedback on services as well as possibly contributing to building trust within the community [9, 13].

There is an ethical obligation to the community to utilize and act upon routinely collected data [8, 14]. Data collected through our surveillance facilitated targeted programmatic interventions and response, including targeted follow-up visits of pregnant women, monitoring of water and sanitation indicators to identify priority areas for latrine desludging and advocacy, and guided timely alert response that was integrated with existing mechanisms including the WHO-led EWARS system.

The CBS, including the alert and response system, was complex and the human resources needed for this surveillance and response mechanism were significant, which had implications for the quantity of trainings and refresher trainings needed, as well as for establishing an effective supervision structure to ensure harmonized methods and adherence to the surveillance protocol. Due to lacking costing data, we were unable to assess the cost-efficiency of the CBS system.

The high acceptability of the CBS by the population might be attributed to surveillance workers and members of the Epi Alert Team and the Water and Sanitation Response Team being recruited from the Rohingya population. Other CBS evaluations have shown that the sense of service to the community, opportunities to increase community ties and improvement in community trust were key motivating factors and increased the acceptability of surveillance workers [15, 16]. The CBS was representative of the population in the catchment area due to its exhaustive character and high household coverage. Despite the human resources complexity and magnitude of the system, the system was flexible enough to allow for the addition of dengue to the diseases of surveillance. During the months under evaluation (May-November 2019) no interruptions of the system occurred. However, it should be noted that between the

onset of the emergency (September 2019) and April 2019 interruptions did occur, and that the system was evaluated when the population movement as well as the CBS had stabilized.

The implementation of the CBS did face its challenges. Regarding the quality of the data collected through the CBS which was the foundation of this evaluation, several limitations have to be noted. Firstly, there were challenges regarding the accuracy of mortality estimates. Surveillance workers were Rohingya and mostly working in the camp where they were residing, which may have facilitated the acceptability of the CBS and accuracy of data collected. In other contexts, the use of CHWs has led to more accurate mortality reporting [8]. On the contrary, anecdotally we learned that people may have underreported deaths to the authorities and NGOs out of fear that their household's food rations would be affected. The underreporting of mortality is a limitation that tends to be noticed in retrospective mortality surveys as well [17, 18]. Secondly, it is possible that households overreported the lack of water and sanitation in the camps, in the hope that more services would be provided, which is a not uncommon limitation in humanitarian needs assessments. Thirdly, at the time of system implementation, the response in Cox's Bazar was at its height with numerous active humanitarian actors. Different organizations were conducting household level interventions, including health promotion activities and surveillance. This may have led to a population fatigued of household visits by different NGOs. The importance of early stakeholder involvement, including community involvement, to avoid duplication and coordinate surveillance activities has been notes by others as well [3, 11, 16, 19]. Finally, as noted in other evaluations of CBS systems, one of the main challenges was the accuracy of the denominator, including the number of households and population covered by the CBS [13]. Due to frequently changing borders of camps, blocks and catchment areas of community leaders, the MSF household numbers had to be frequently updated by surveillance workers, which probably negatively impacted the accuracy of the denominators. The lack of reliable denominators impaired our ability to calculate mortality and birth rates, reliable track population movements and calculate disease incidence.

Different stages of an emergency demand a different level of exhaustiveness of a CBS to fulfill different needs and depending on levels of health facilities access. In the initial phase it is very important to make sure that cases are not missed, and early referral procedures are in place to avoid undetected outbreaks. The fact that detected disease trends were similar and cholera cases were picked-up by health facility based surveillance as well as CBS, might indicate limited additional value of the CBS in a dense and stable setting such as Cox's Bazar. Instead, a passive community-event-based surveillance mechanism combined with health facility-based surveillance could be more appropriate. This would require fewer resources for data collection, still allowing for morbidity trends monitoring and including an early warning of important public health events [4].

## Supporting information

**S1 File. AWD case investigation form (exported from KoboCollect).**
(TIFF)

## Author Contributions

**Conceptualization:** Elburg Van Boetzelaer, Samiur Chowdhury, Berhe Etsay, Abu Faruque, Annick Lenglet, Anna Kuehne, Isidro Carrion-Martin, Patrick Keating, Donald Sonne Kazungu, Maria Verdecchia.

**Data curation:** Elburg Van Boetzelaer, Samiur Chowdhury, Berhe Etsay.

**Formal analysis:** Elburg Van Boetzelaer, Maria Verdecchia.

**Methodology:** Elburg Van Boetzelaer, Samiur Chowdhury, Berhe Etsay, Annick Lenglet, Anna Kuehne, Isidro Carrion-Martin, Patrick Keating, Donald Sonne Kazungu, Maria Verdecchia.

**Supervision:** Elburg Van Boetzelaer, Samiur Chowdhury, Donald Sonne Kazungu, Maria Verdecchia.

**Validation:** Elburg Van Boetzelaer.

**Visualization:** Elburg Van Boetzelaer, Samiur Chowdhury, Jorieke Vyncke, Maria Verdecchia.

**Writing – original draft:** Elburg Van Boetzelaer, Maria Verdecchia.

**Writing – review & editing:** Samiur Chowdhury, Berhe Etsay, Abu Faruque, Annick Lenglet, Anna Kuehne, Isidro Carrion-Martin, Patrick Keating, Martins Dada, Donald Sonne Kazungu.

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
