## [Decision Letter · Decision Letter 0]

10 Aug 2020

PONE-D-20-14072

Evaluation of Community Based Surveillance in the Rohinya refugee camps in Cox's Bazar, Bangladesh, 2019

PLOS ONE

Dear Dr. van Boetzelaer,

Thank you for submitting your manuscript to PLOS ONE. After careful consideration, we feel that it has merit but does not fully meet PLOS ONE’s publication criteria as it currently stands. Therefore, we invite you to submit a revised version of the manuscript that addresses the points raised during the review process.

We look forward to receiving your revised manuscript.

Kind regards,

Jennifer Yourkavitch

Academic Editor

PLOS ONE

Additional Editor Comments:

Please address all reviewer comments. Please also indicate if data used in the analysis are available, i.e., the underlying data from the surveillance system are not in the manuscript or supplemental files.

Journal Requirements:

2. We note that Figure 2 in your submission contain map images which may be copyrighted. All PLOS content is published under the Creative Commons Attribution License (CC BY 4.0), which means that the manuscript, images, and Supporting Information files will be freely available online, and any third party is permitted to access, download, copy, distribute, and use these materials in any way, even commercially, with proper attribution. For these reasons, we cannot publish previously copyrighted maps or satellite images created using proprietary data, such as Google software (Google Maps, Street View, and Earth). For more information, see our copyright guidelines: http://journals.plos.org/plosone/s/licenses-and-copyright.

2.1.    You may seek permission from the original copyright holder of Figure 2 to publish the content specifically under the CC BY 4.0 license.

2.2.    If you are unable to obtain permission from the original copyright holder to publish these figures under the CC BY 4.0 license or if the copyright holder’s requirements are incompatible with the CC BY 4.0 license, please either i) remove the figure or ii) supply a replacement figure that complies with the CC BY 4.0 license. Please check copyright information on all replacement figures and update the figure caption with source information. If applicable, please specify in the figure caption text when a figure is similar but not identical to the original image and is therefore for illustrative purposes only.

Reviewers' comments:

Reviewer's Responses to Questions

**Comments to the Author**

1. Is the manuscript technically sound, and do the data support the conclusions?

Reviewer #1: No

Reviewer #2: Yes

2. Has the statistical analysis been performed appropriately and rigorously? 

Reviewer #1: N/A

Reviewer #2: Yes

3. Have the authors made all data underlying the findings in their manuscript fully available?

Reviewer #1: Yes

Reviewer #2: No

4. Is the manuscript presented in an intelligible fashion and written in standard English?

Reviewer #1: Yes

Reviewer #2: Yes

5. Review Comments to the Author

Reviewer #1: Dear Authors,

Congratulations for your hard work in setting up and operating a large community-based surveillance system in this emergency setting. Due to its scale and its context, I strongly believe the description of this system in the literature would be of important interest.

I read your manuscript with attention, but I still struggle with some specifics of how the system was operated, but mainly with its results. I would like to understand how many cases were detected because of the household visits, or how many cases were first detected through the household visits in comparison with the detected cases in the routine healthcare facility system. At the moment, it is still not clear to me.

Also, I consider this paper rather as a description of the system and its results, than as a proper dedicated evaluation of the system which would have needed the collection of dedicated data to inform pre-defined evaluation indicators. I don’t think it is a problem, a description is already of great interest, I would just suggest replacing all instances of “evaluation” by “description”.

Please find below detailed comments and proposed modifications for your consideration, I hope they may be useful.

Best regards,

Title:

* Consider the replacement of the term “Evaluation” by “Description”: i.e. “Description of Community Based Surveillance […]”, indeed the paper provides a description of the structure and several attributes of the surveillance system put in place and of its results, rather than a proper dedicated evaluation with pre-planned methodology and use of dedicated evaluation indicators.

Abstract:

* Line 25: replace “evaluated” by “described” as it seems no additional data was collected to specifically “evaluate” the performance of the CBS system in comparison with a set of pre-defined evaluation indicators.

* Line 30: not clear if these cholera cases were identified by the CBS system, by the healthcare facilities, or by both.

* Line 37: there is a need to know how many cases were detected solely by health facility surveillance and how many through CBS to be able to make a conclusion on the value of the CBS system, see below comments for additional details.

Background:

* Lines 55-57: from references 3 to 7 only reference 5 relates with the assertion that “[CBS] has been shown to facilitate early detection of epidemic prone diseases […]." The other references relate either to systems focused on mortality and nutritional status (3,4) or healthcare facility based systems (6,7). Thus, please remove the incorrect references to the claim and consider rewording it as “It may facilitate” instead of “It has been shown to facilitate”.

* Line 73: consider removing “evaluate” from “We describe and evaluate”.

Figure 1:

* Consider introducing the figure later in the methods section once some concepts such as EWARS tool have been described.

* Define all acronyms in the footnote, e.g. AWD: acute watery diarrhoea; WHO-led JAT investigation.

* On the arrow before "referral of suspected case to nearest health facility" specify that it is done if it has been considered a suspect case by the visiting team.

* Discrepancy with information lines 103-104 where all suspect cases were referred by the surveillance worker to the healthcare facility OR the figure 1 is misleading (lines 103-104: "Surveillance workers referred all suspected cases of disease to the nearest health facility and recorded the household number to inform further case investigations by the Epi Alert Team or Medical Response Team”).

* Consider rewording "Upload case information suspected case on EWARS (weekly)" to "Weekly upload of case information on EWARS".

* Did you have an RDT for AFP? I guess this was only for cholera and hepatitis E, to be corrected.

Methods:

* Provide a dedicated section on the surveillance workers: what were their recruitment criteria, which training was provided, for how long, on which topics, which refreshers?

* It is not clear which data you used for the descriptions provided in the results section, for example which source of data you used for the number of cases identified by health care facilities? Please describe which data you used and how you analysed it.

* Line 87: could the comparison with the camps without CBS have been considered?

* Lines 87-88: “Each surveillance cycle encompassed four weeks” can be removed here as it will be described later and we don’t know yet what you mean by “surveillance cycle”.

* Lines 93-97: Consider rewording ￹"[…] data were collected on active case finding for specific epidemic prone diseases" to "[…] data were collected by active case finding of specific epidemic prone diseases, [...]". See also comment lines 106-111.

* Lines 103-104: discrepancy with figure 1, see comment above in Figure 1.

* Lines 106-111: merge with lines 93-97, but it would be even better to remove these paragraphs and replace them by a table with a comprehensive list of the collected variables, or to expand figure 3 to include the collected variables, or to provide a copy of the collecting form. You can also consider moving this list of collected data to the section currently named “Data analysis and reporting” (lines 138-145).

* Lines 123-124: specify the modalities of the household visits: was a surveillance worker alone during the visits? Or they were two surveillance workers per visit?

* Lines 128-129: How many surveillance workers per team leader?

* Lines 138-145: Consider renaming the section “Data analysis and reporting” to “Data entry and data analysis” and put it after the section “Alert and Response” to ease its understanding (for example lines 143-145 there was no prior mention of data collected by the Epi Alert Team and the Medical Response teams).

* Lines 139-140: It is not clear which data was entered in an electronic system and how. When you list the variables collected (see previous comments), can you specify if data has been entered in an electronic system (such as Excel spreadsheets), how (specific data entry mask or not), if it was done through a single or double entry, and if it was done through dedicated data entry staff or other staff on top of their other duties.

* Lines143-145: please specify which data is collected by the Epi Alert Team and Medical Response teams and how, you can provide the link to the supplementary material 1.

* Lines 155-156: need to quickly describe the EWARS system, or at least put the current reference 10.

* Lines 180-181: in Figure 1 it is mentioned that case-based data (line lists) were entered in the EWARS system weekly. Did healthcare facilities perform data entry in Excel in addition to the entry in EWARS? If yes, by whom and please discuss why.

* Lines 190-191: “ Firstly, the epidemiologists reported the probable cases of Cholera

and Hepatitis E into the EWARS mechanism”, discrepancy with figure 1 or unclear statement: in figure 1 such action is performed if probable case of cholera and hepatitis E but also of AFP and clusters of AJS and AWD.

* Line 194: “icddr,b” acronym, can you modify this acronym to remove the comma in it and put some capital letters to help the reader?

Table 1:

* I understand you used the same case definitions at healthcare facility level and for the community health workers.

* Yet, these are very difficult case definitions for community workers , e.g. terms such as "anorexia", "sclera", "focal weakness or paralysis" or paraclinic criteria: e.g. "leukopenia".

* Please specify why dedicated definitions for community workers were not used while available in international guidance (see list of existing case definitions for CBS in the "Data collection" section of the following document https://www.ncbi.nlm.nih.gov/pmc/articles/PMC6461245/bin/pone.0215278.s002.pdf from https://dx.doi.org/10.1371%2Fjournal.pone.0215278.

* Please discuss your choice of using the same case definitions for CBS in healthcare facilities in the discussion section.

Table 3:

* The reference is wrong, the CDC guidance has been published in 2001, not in 2012.

* Please put the attributes and “measurement” in the same order as in the result section and remove any attribute/measurement you don’t describe in the result section.

* Please put the data you used for each measurement instead of excerpts from the CDC guidance.

* Sensitivity attribute: the “verification of each suspected case identified by the CBS at the health facility level” would not help to assess the sensitivity of the system; the only way to assess this sensitivity would have been to first undertake a dedicated study to detect all cases really occurring in the area of interest (which, I imagine, would not have been feasible).

Results:

* Provide a short description of the surveillance workers: numbers, demographics (age, sex), literacy level.

* Do you have data on the number of cases identified by healthcare facilities without referral by CBS, and the number of cases referred by CBS?

* Lines 233-234, Figures 4 to 7: this is very unclear:

* If I understand well, your Epi Alert Team is sent to the household to check the suspect case reported by the community health worker. If I am correct, this "Epi Alert Team" is then part of your CBS system and it makes no sense to present a "Epi Alert Team" curve in addition to the "CBS" curve. "CBS curve" that should be the current "Epi Alert Team curve".

* How can the "Health facilities" curve be below the "Epi Alert Team" or "CBS" curves while all CBS suspect cases were expected to be referred to the health facility and thus be registered there? Please provide more information regarding the data collection and analysis process of these "Health facilities suspect cases".

* Lines 245-255: you should present here the usefulness of the CBS system in addition to the routine healthcare facility system. Do you have any data regarding how many cases were referred to the healthcare facilities because of the CBS system? (And how many were identified in the healthcare facilities without being referred by the CBS system).

* Lines 267-269: “The CBS supplemented the health facility based surveillance as it helped to identify suspected cases of disease early, and to ensure timely referral of suspected cases to health facilities for

diagnostics and treatment.” I don’t see any result supporting this claim, please provide some data to support it, or you need to remove it.

* Line 271: Timeliness: the main indicator would have been the delay between onset of symptoms and identification to see if CBS managed to identify suspect cases earlier.

* Lines 273-275: please rephrase to "the main difference is the delay between the identification of a suspected case and the notification of the epidemiologist and Epi Alert Team or Medical Response Team through health facility based surveillance versus through CBS".

* Lines 288-289: “While this surveillance system generated data that informed programming and enabled early detection of epidemic prone disease […]”, please provide data to support this claim, or remove it.

* Lines 292-298: Please put this section regarding training in the methods.

Table 5:

* Again it is not clear what the referral process was. Any suspect case by community surveillance workers were referred for case management even if the case was not verified by the EPI alert team? Here, all the suspect cases have a public health action performed even if they are discarded by the Epi Alert team or the Medical Response Team. Please explain.

Table 6:

* Why is there only data on AWD cases?

Discussion:

* Lines 325-327: “ This evaluation shows that the CBS system and the alert and response mechanism allowed for timely detection and response to cases of epidemic prone diseases”: which results support this claim?

* Lines 346-348: the reason why you couldn’t assess the sensitivity of the system was that you couldn’t know the real occurrence of the cases in the area, not that you couldn’t check that the suspect cases were true positives.

* Lines 350-352: the fact that the population under surveillance is different between the CBS system and the healthcare facility system is very important and should be emphasized in the methods section and taken into consideration in the results.

* Lines 411-416: Please specify for which indicator under surveillance the lack of proper denominator was a problem, and which consequences it had. For example, regarding the detection of acute infectious diseases, this was not a problem as the objective was to early detect and early respond to first cases to avoid an outbreak.

* Lines 420-426: while I agree with the general conclusion, I don't see in this paper clear results regarding the additional number of cases that were identified by CBS in comparison to the healthcare facility system.

Reviewer #2: This is a clearly written article with technically rigorous rationale, methods, findings, and conclusions.

In the data quality/validity section, I suggest the authors add more explanation to help the reader interpret the graphs, which is (if my interpretation is correct): that, in addition to the trends being similar, that the CBS consistently identified more cases. But when verified by the Epi Alert Team or Medical Response Team the numbers of suspected cases were similar to that of the health facility system. The possible exception may be AJS, but those numbers were fewer relative to other suspected cases.

I appreciate the authors and the journal's willingness to publish not-positive findings. That said, there could have been a number of positive externalities of the CBS system that are not captured by the 'number of suspected' cases data. The result of the system in terms of referral of CHWs, water and sanitation, and ability of health services to respond to community needs/opinions could have a overall positive health effect. Based on their knowledge of the situation, perhaps the authors have a hypothesis of how to re-organize services and/or better coordinate services with other actors based on this experience, one that is not as focused on the CBS system.

Minor comment: on lines 379-380, suggest the authors specify which data were lacking (assume it was costing data?). On the other hand, given the findings, one might conclude that it is not necessary to conduct any costing study.

6. PLOS authors have the option to publish the peer review history of their article (what does this mean?). If published, this will include your full peer review and any attached files.

Reviewer #1: **Yes: **Jose Guerra

Reviewer #2: No

---

## [Author Response · Author response to Decision Letter 0]

13 Oct 2020

Amsterdam, 8 September 2020

Dear Editor,

We would like to thank you for the consideration of our manuscript ‘Description of Community Based Surveillance in the Rohingya refugee camps in Cox’s Bazar, Bangladesh, 2019’ for publication in PlosOne. In addition, we would like to thank both reviewers very much for taking the time to review our draft manuscript thoroughly and share such insightful comments. We have addressed all comments, kindly find our responses below.

Kind regards, on behalf of the authors,

Elburg van Boetzelaer (corresponding author)

Reviewer #1: 

Dear Authors,

Congratulations for your hard work in setting up and operating a large community-based surveillance system in this emergency setting. Due to its scale and its context, I strongly believe the description of this system in the literature would be of important interest.

I read your manuscript with attention, but I still struggle with some specifics of how the system was operated, but mainly with its results. I would like to understand how many cases were detected because of the household visits, or how many cases were first detected through the household visits in comparison with the detected cases in the routine healthcare facility system. At the moment, it is still not clear to me.

Also, I consider this paper rather as a description of the system and its results, than as a proper dedicated evaluation of the system which would have needed the collection of dedicated data to inform pre-defined evaluation indicators. I don’t think it is a problem, a description is already of great interest, I would just suggest replacing all instances of “evaluation” by “description”.

Please find below detailed comments and proposed modifications for your consideration, I hope they may be useful.

Best regards,

Title:

* Consider the replacement of the term “Evaluation” by “Description”: i.e. “Description of Community Based Surveillance […]”, indeed the paper provides a description of the structure and several attributes of the surveillance system put in place and of its results, rather than a proper dedicated evaluation with pre-planned methodology and use of dedicated evaluation indicators. Thank you for the feedback. We have considered replacing ‘evaluation’ with ‘description’, however we believe that the manuscript we wrote on the assessment we did of the Community Based Surveillance system that was implemented by MSF in Cox’s Bazar does meet the definition of an evaluation. While some parts of the manuscript (such as the case definitions, data flow, roles and responsibilities, SOPs) and surveillance system attributes are descriptive in nature, there are other parts that are more evaluative in nature. We believe that for example the comparison of the Community Based Surveillance and the Health Facility surveillance systems, as well as the assessment of the PPV and timeliness are inherent attributes to surveillance systems that can only be calculated if a system is evaluated. Moreover, we did use a pre-planned methodology to conceptualize this evaluation and for example considered different indicators to evaluate the performance of the CBS system at its conception (for example, we decided to collect additional data on the number of alerts meeting case definitions so that we could evaluate the system’s performance). Therefore, we would strongly prefer to consider this assessment as an evaluation, with descriptive components.

Abstract:

* Line 25: replace “evaluated” by “described” as it seems no additional data was collected to specifically “evaluate” the performance of the CBS system in comparison with a set of pre-defined evaluation indicators. Please see above.

* Line 30: not clear if these cholera cases were identified by the CBS system, by the healthcare facilities, or by both. By the CBS and health facility surveillance, clarification added.

* Line 37: there is a need to know how many cases were detected solely by health facility surveillance and how many through CBS to be able to make a conclusion on the value of the CBS system, see below comments for additional details. Unfortunately we were not able to track CBS referrals because there were many health facilities in the catchment area, only two of them operated by MSF OCA. This means that what we do know is that surveillance workers referred suspected cases to the ‘nearest health facility’ (whether that was an MSF facility or not). What we do not know is how many of those referrals actually went to a health facility, and of the patients that were seen at the MSF facility how many of those were referred to the facility by the surveillance workers (we were unable for example to successfully implement a ‘referral slip’ that would be handed out by surveillance workers to suspected cases. We tried to give referral slips to suspected Diphtheria cases as the MSF facility was the only treatment center for Diphtheria in the entire megacamp, but without much success). We have added a sentence on this for clarification in the abstract.

Background:

* Lines 55-57: from references 3 to 7 only reference 5 relates with the assertion that “[CBS] has been shown to facilitate early detection of epidemic prone diseases […]." The other references relate either to systems focused on mortality and nutritional status (3,4) or healthcare facility based systems (6,7). Thus, please remove the incorrect references to the claim and consider rewording it as “It may facilitate” instead of “It has been shown to facilitate”. We have removed and reworded as suggested.

* Line 73: consider removing “evaluate” from “We describe and evaluate”. Please see above.

Figure 1:

* Consider introducing the figure later in the methods section once some concepts such as EWARS tool have been described. We have moved the figure with the overview of the surveillance system to the end of the ‘methods’ section, line 204, as suggested.

* Define all acronyms in the footnote, e.g. AWD: acute watery diarrhoea; WHO-led JAT investigation. We have added as suggested.

* On the arrow before "referral of suspected case to nearest health facility" specify that it is done if it has been considered a suspect case by the visiting team. We have added as suggested.

* Discrepancy with information lines 103-104 where all suspect cases were referred by the surveillance worker to the healthcare facility OR the figure 1 is misleading (lines 103-104: "Surveillance workers referred all suspected cases of disease to the nearest health facility and recorded the household number to inform further case investigations by the Epi Alert Team or Medical Response Team”). We have added a text box to show that the surveillance workers AND the epi alert team & medical response team refer suspected cases to the nearest health facility (see later comment for explanation).

* Consider rewording "Upload case information suspected case on EWARS (weekly)" to "Weekly upload of case information on EWARS". We have reworded as suggested.

* Did you have an RDT for AFP? I guess this was only for cholera and hepatitis E, to be corrected. We have corrected as suggested, no RDT for AFP.

Methods:

* Provide a dedicated section on the surveillance workers: what were their recruitment criteria, which training was provided, for how long, on which topics, which refreshers? We have moved the paragraph on training from the ‘results – simplicity’ section to the methods section as suggested.

* It is not clear which data you used for the descriptions provided in the results section, for example which source of data you used for the number of cases identified by health care facilities? Please describe which data you used and how you analysed it. We have added the data sources to Table 3.

* Line 87: could the comparison with the camps without CBS have been considered? It would have been interesting, but MSF OCA implemented CBS in their entire catchment area (13 sub-camps). Therefore, we did not have access to health facility data outside of our catchment area to make any comparisons. In addition, in other sub-camps outside of the MSF OCA catchment area, other NGOs were implementing different forms of CBS (some indicator based, others event based). It would be very interesting to compare different parts of the megacamp and different surveillance mechanisms, but unfortunately that exceeds the scope of this manuscript.

* Lines 87-88: “Each surveillance cycle encompassed four weeks” can be removed here as it will be described later and we don’t know yet what you mean by “surveillance cycle”. We have removed as suggested.

* Lines 93-97: Consider rewording ￹"[…] data were collected on active case finding for specific epidemic prone diseases" to "[…] data were collected by active case finding of specific epidemic prone diseases, [...]". See also comment lines 106-111. We have reworded as suggested.

* Lines 103-104: discrepancy with figure 1, see comment above in Figure 1. We have adapted Figure 1.

* Lines 106-111: merge with lines 93-97, but it would be even better to remove these paragraphs and replace them by a table with a comprehensive list of the collected variables, or to expand figure 3 to include the collected variables, or to provide a copy of the collecting form. You can also consider moving this list of collected data to the section currently named “Data analysis and reporting” (lines 138-145). We have expanded Figure 3 to include the variables that were collected. And removed L 93-97 and L106-111.

* Lines 123-124: specify the modalities of the household visits: was a surveillance worker alone during the visits? Or they were two surveillance workers per visit? We have added that the households were visited by ‘a surveillance worker’ (single).

* Lines 128-129: How many surveillance workers per team leader? The team leaders were leading multidisciplinary outreach teams, including 4 surveillance workers, 2 health promotors, 2 traditional birth attendants, 2 women’s health promotors. Added that each team leader supervised 4 surveillance workers to manuscript.

* Lines 138-145: Consider renaming the section “Data analysis and reporting” to “Data entry and data analysis” and put it after the section “Alert and Response” to ease its understanding (for example lines 143-145 there was no prior mention of data collected by the Epi Alert Team and the Medical Response teams). We have renamed and moved as suggested.

* Lines 139-140: It is not clear which data was entered in an electronic system and how. When you list the variables collected (see previous comments), can you specify if data has been entered in an electronic system (such as Excel spreadsheets), how (specific data entry mask or not), if it was done through a single or double entry, and if it was done through dedicated data entry staff or other staff on top of their other duties. We have rewritten the paragraph on ‘data entry and data analysis’ in the hope that it is now clearer. Surveillance workers collected data on paper based forms, that were subsequently entered into and Microsoft Excel data base by two full time data encoders. Data on suspected morbidities, verbal autopsies and water and sanitation indicators were entered real time by the Epi Alert Team and Medical Response Team while they were at the house of the suspected case on tablets/smart phones.

* Lines143-145: please specify which data is collected by the Epi Alert Team and Medical Response teams and how, you can provide the link to the supplementary material 1. We have added as suggested.

* Lines 155-156: need to quickly describe the EWARS system, or at least put the current reference 10. We have added reference 10.

* Lines 180-181: in Figure 1 it is mentioned that case-based data (line lists) were entered in the EWARS system weekly. Did healthcare facilities perform data entry in Excel in addition to the entry in EWARS? If yes, by whom and please discuss why. Yes, data encoders enter data into EWARS as well as into MSF’s HIS system, which at the time was largely excel based (although it is now shifting to DHIS2). Added to ‘data collection, analysis & reporting’ section.

* Lines 190-191: “ Firstly, the epidemiologists reported the probable cases of Cholera

and Hepatitis E into the EWARS mechanism”, discrepancy with figure 1 or unclear statement: in figure 1 such action is performed if probable case of cholera and hepatitis E but also of AFP and clusters of AJS and AWD. Clarified in the text. “Firstly, the epidemiologists reported probable cases of Cholera, Hepatitis E, AJS as well as clusters of AJS and AWD into the EWARS mechanism. This activated the WHO-led Joint Assessment Team (JAT) who conducted a case investigation.”

* Line 194: “icddr,b” acronym, can you modify this acronym to remove the comma in it and put some capital letters to help the reader? We do not think we can adjust this, as the official acronym of the International Centre for Diarrhoeal Disease Research in Bangladesh is actually icddr,b: https://www.icddrb.org/

Table 1:

* I understand you used the same case definitions at healthcare facility level and for the community health workers. Apologies, this was a translation/language issue and Table 1 has been adapted accordingly.

* Yet, these are very difficult case definitions for community workers , e.g. terms such as "anorexia", "sclera", "focal weakness or paralysis" or paraclinic criteria: e.g. "leukopenia". See above.

* Please specify why dedicated definitions for community workers were not used while available in international guidance (see list of existing case definitions for CBS in the "Data collection" section of the following document https://www.ncbi.nlm.nih.gov/pmc/articles/PMC6461245/bin/pone.0215278.s002.pdf from https://dx.doi.org/10.1371%2Fjournal.pone.0215278.

* Please discuss your choice of using the same case definitions for CBS in healthcare facilities in the discussion section. See above.

Table 3:

* The reference is wrong, the CDC guidance has been published in 2001, not in 2012. We have corrected as suggested.

* Please put the attributes and “measurement” in the same order as in the result section and remove any attribute/measurement you don’t describe in the result section. We have reordered as suggested.

* Please put the data you used for each measurement instead of excerpts from the CDC guidance. We have added a column to Table 3 with the data sources.

* Sensitivity attribute: the “verification of each suspected case identified by the CBS at the health facility level” would not help to assess the sensitivity of the system; the only way to assess this sensitivity would have been to first undertake a dedicated study to detect all cases really occurring in the area of interest (which, I imagine, would not have been feasible). Correct, therefore ‘sensitivity’ was not included in this description.

Results:

* Provide a short description of the surveillance workers: numbers, demographics (age, sex), literacy level. We have added a sentence ‘There were 140 surveillance workers, of which one female surveillance worker. All surveillance workers were literate.’ Unfortunately we do not have any data on the ages of the surveillance workers nor specific data on their education levels (except for the fact that they were all literate). The surveillance workers were predominantly male as it is considered culturally inappropriate for women to work.

* Do you have data on the number of cases identified by healthcare facilities without referral by CBS, and the number of cases referred by CBS? Unfortunately we were not able to track CBS referrals because there were many health facilities in the catchment area, only two of them operated by MSF OCA. This means that what we do know is that surveillance workers referred suspected cases to the ‘nearest health facility’ (whether that was an MSF facility or not). What we do not know is how many of those referrals actually went to a health facility, and of the patients that were seen at the MSF facility how many of those were referred to the facility by the surveillance workers (we were unable for example to succesfully implement a ‘referral slip’ that would be handed out by surveillance workers to suspected cases. We tried to give referral slips to suspected Diphtheria cases as the MSF facility was the only treatment center for Diphtheria in the entire megacamp, but without much success).

* Lines 233-234, Figures 4 to 7: this is very unclear:

* If I understand well, your Epi Alert Team is sent to the household to check the suspect case reported by the community health worker. If I am correct, this "Epi Alert Team" is then part of your CBS system and it makes no sense to present a "Epi Alert Team" curve in addition to the "CBS" curve. "CBS curve" that should be the current "Epi Alert Team curve". Well noted, in fact, the ‘CBS curve’ contains the suspected cases identified by the surveillance workers. The ‘Epi Alert Team curve’ contains the suspected cases that were verified as such by the Epi Alert Team. So, the surveillance workers identified a suspected case, then the Epi Alert Team went to the household to verify that this person in fact met the case definition of a suspected case. That way, we could verify the quality of data collected by the surveillance workers. Therefore, we will reword the ‘CBS curve’ to be ‘Surveillance worker curve’ in figure 4-7. We have also added a sentence to the paragraph to hopefully clarify this.

* How can the "Health facilities" curve be below the "Epi Alert Team" or "CBS" curves while all CBS suspect cases were expected to be referred to the health facility and thus be registered there? Please provide more information regarding the data collection and analysis process of these "Health facilities suspect cases". All suspected cases that were identified by the surveillance workers were referred to the nearest health facility. Then, the next day when the Epi Alert Team came for the follow up/verification visit, they would remind the suspected case of the referral to the health facility (anecdotally we know that many of the initial referrals are not completed due to lack of trust towards health facilities). So in essence, suspected were referred by the surveillance workers and reminded to ‘please go to the health facility for assessment and care’ by the Epi Alert Team. However in the catchment area there are many health facilities, only two of them operated by MSF. Therefore, the number of suspected cases at the MSF health facilities is lower than those identified by the surveillance workers and verified by the Epi Alert Team. We have added a sentence to the narrative to hopefully clarify.

* Lines 245-255: you should present here the usefulness of the CBS system in addition to the routine healthcare facility system. Do you have any data regarding how many cases were referred to the healthcare facilities because of the CBS system? (And how many were identified in the healthcare facilities without being referred by the CBS system). All suspected cases that were identified by the surveillance workers were referred to the nearest health facility (not necessarily an MSF OCA health facility), and then the next day the Epi Alert Team verified the suspected cases and reminded them of the referral (if they had not visited the nearest health facility yet). Unfortunately we were unable to track whether referrals actually arrived at health facilities and whether suspected cases at MSF OCA health facilities were referred by the CBS.

* Lines 267-269: “The CBS supplemented the health facility based surveillance as it helped to identify suspected cases of disease early, and to ensure timely referral of suspected cases to health facilities for

diagnostics and treatment.” I don’t see any result supporting this claim, please provide some data to support it, or you need to remove it. We have removed this sentence.

* Line 271: Timeliness: the main indicator would have been the delay between onset of symptoms and identification to see if CBS managed to identify suspect cases earlier. Unfortunately, we do not have reliable data to analyse this.

* Lines 273-275: please rephrase to "the main difference is the delay between the identification of a suspected case and the notification of the epidemiologist and Epi Alert Team or Medical Response Team through health facility based surveillance versus through CBS". We have rephrased as suggested.

* Lines 288-289: “While this surveillance system generated data that informed programming and enabled early detection of epidemic prone disease […]”, please provide data to support this claim, or remove it. Thank you for the feedback. We considered whether we provided data to support this claim, and we believe that we did provide sufficient data. For example, we showed that the two AWD clusters that were identified by surveillance workers triggered a multidisciplinary response mechanism, including a water and sanitation intervention, active case finding, and a Join Assessment Team investigation by the WHO. In addition, we showed in figures 4-7 that the CBS identified many more suspected cases of outbreak prone diseases than the health facility surveillance did.

* Lines 292-298: Please put this section regarding training in the methods. We have moved the section to ‘methods’ as suggested.

Table 5:

* Again it is not clear what the referral process was. Any suspect case by community surveillance workers were referred for case management even if the case was not verified by the EPI alert team? Here, all the suspect cases have a public health action performed even if they are discarded by the Epi Alert team or the Medical Response Team. Please explain. Correct, see above, surveillance workers referred all suspected cases. Then the next day, the Epi Alert Team verified the suspected case and reminder the case of their referral to the nearest health facility, if they had not visited the health facility yet.

Table 6:

* Why is there only data on AWD cases? We included only the AWD clusters as identified by the CBS and 21 RDT+ cholera cases as identified by the health facility surveillance in our assessment of the timeliness as these were the only ones during the period under evaluation that triggered a ‘full fledged’ alert response (deployment of the Epi Alert Team for contact tracing and active case finding, deployment of the Water & Sanitation Response Team for latrine cleaning, bucket chlorination and hygiene promotion sessions, deployment of WHO’s JAT for investigation and icddr,b for sample taking). Therefore the clusters of AWD and RDT+ Cholera cases were the most appropriate to assess the timeliness of the response mechanism.

Discussion:

* Lines 325-327: “ This evaluation shows that the CBS system and the alert and response mechanism allowed for timely detection and response to cases of epidemic prone diseases”: which results support this claim? We refer here to the assessment of timeliness (lines 277 and further) and Table 6, in which we have assessed the timeliness of the different steps or phases of the CBS system.

* Lines 346-348: the reason why you couldn’t assess the sensitivity of the system was that you couldn’t know the real occurrence of the cases in the area, not that you couldn’t check that the suspect cases were true positives. We have added as suggested.

* Lines 350-352: the fact that the population under surveillance is different between the CBS system and the healthcare facility system is very important and should be emphasized in the methods section and taken into consideration in the results. We have added to methods – passive health facility surveillance ‘population and diseases under surveillance’.

* Lines 411-416: Please specify for which indicator under surveillance the lack of proper denominator was a problem, and which consequences it had. For example, regarding the detection of acute infectious diseases, this was not a problem as the objective was to early detect and early respond to first cases to avoid an outbreak. We have added a sentence: ‘The lack of reliable denominators impaired our ability to calculate mortality and birth rates, reliable track population movements and calculate disease incidence.’

* Lines 420-426: while I agree with the general conclusion, I don't see in this paper clear results regarding the additional number of cases that were identified by CBS in comparison to the healthcare facility system. Figure 4-7 show this.

Reviewer #2: This is a clearly written article with technically rigorous rationale, methods, findings, and conclusions.

In the data quality/validity section, I suggest the authors add more explanation to help the reader interpret the graphs, which is (if my interpretation is correct): that, in addition to the trends being similar, that the CBS consistently identified more cases. But when verified by the Epi Alert Team or Medical Response Team the numbers of suspected cases were similar to that of the health facility system. The possible exception may be AJS, but those numbers were fewer relative to other suspected cases. Correct interpretation, we have added a bit more to the narrative: “The trend comparison per disease show that surveillance workers identified the most suspected cases, of which a proportion was verified by either the Epi Alert Team or the Medical Response Team as suspected cases (‘true positives’) and a proportion was disregarded as ‘false positives’. At the two MSF health facilities, which had a different catchment population than the CBS, similar trends were detected albeit with lower suspected case numbers.”

I appreciate the authors and the journal's willingness to publish not-positive findings. That said, there could have been a number of positive externalities of the CBS system that are not captured by the 'number of suspected' cases data. The result of the system in terms of referral of CHWs, water and sanitation, and ability of health services to respond to community needs/opinions could have a overall positive health effect. Based on their knowledge of the situation, perhaps the authors have a hypothesis of how to re-organize services and/or better coordinate services with other actors based on this experience, one that is not as focused on the CBS system. The overall positive effect that the CBS system could have had in the camp is possible but not easily measurable. Coordination with other actors working in the camp is rather an important and essential issue that certainly needs to be addressed but it goes beyond the scope of this paper. 

Minor comment: on lines 379-380, suggest the authors specify which data were lacking (assume it was costing data?). On the other hand, given the findings, one might conclude that it is not necessary to conduct any costing study. We have added ‘costing data’.

---

## [Decision Letter · Decision Letter 1]

16 Nov 2020

PONE-D-20-14072R1

Evaluation of Community Based Surveillance in the Rohinya refugee camps in Cox's Bazar, Bangladesh, 2019

PLOS ONE

Dear Dr. van Boetzelaer,

Thank you for submitting your manuscript to PLOS ONE. After careful consideration, we feel that it has merit but does not fully meet PLOS ONE’s publication criteria as it currently stands. Therefore, we invite you to submit a revised version of the manuscript that addresses the points raised during the review process.

We look forward to receiving your revised manuscript.

Kind regards,

Jennifer Yourkavitch

Academic Editor

PLOS ONE

Additional Editor Comments (if provided):

Thank you for your attention to the reviewers' comments. There are a few more issues to address.

Reviewers' comments:

Reviewer's Responses to Questions

**Comments to the Author**

1. If the authors have adequately addressed your comments raised in a previous round of review and you feel that this manuscript is now acceptable for publication, you may indicate that here to bypass the “Comments to the Author” section, enter your conflict of interest statement in the “Confidential to Editor” section, and submit your "Accept" recommendation.

Reviewer #1: (No Response)

Reviewer #2: All comments have been addressed

2. Is the manuscript technically sound, and do the data support the conclusions?

Reviewer #1: Partly

Reviewer #2: Yes

3. Has the statistical analysis been performed appropriately and rigorously? 

Reviewer #1: Yes

Reviewer #2: Yes

4. Have the authors made all data underlying the findings in their manuscript fully available?

Reviewer #1: Yes

Reviewer #2: Yes

5. Is the manuscript presented in an intelligible fashion and written in standard English?

Reviewer #1: Yes

Reviewer #2: Yes

6. Review Comments to the Author

Reviewer #1: Dear Authors,

I think the paper is much clearer now, thank you for your commitment.

I believe this paper should be published, this is an important experience which involved a lot of resources including hundreds of committed colleagues during an extended period of time.

I just have two major issues that I think should be taken into consideration before publication and few minor comments.

**Major issues:**

* **Data quality / validity evaluation attribute**:

* In Table 3, the measurement modality for the "Data quality / validity evaluation attribute" is stated as "Comparison of trends of health facility surveillance (gold standards) and community based surveillance trends".

* The fact is that **you can't compare these trends as they don't have the same population under surveillance**.

* You correctly say it in the discussion lines 357-360: "While the trends of reported suspected cases from all the components of the surveillance system were similar, and could be an indicator of data quality and validity, **it must be noted that the catchment population of the CBS and health facility surveillance differed** as the health facilities were also visited by patients who did not live in the CBS catchment area."

* In any case, **it is misleading to evaluate this "Data quality / validity" attribute, as you have no data to do it.**

* Lines 240-249: when you compare the "Surveillance worker curve" and the "Epi alert team curve", you are not evaluating the "data quality/validity" but the evolution of the PPV. The assertion "The trend comparison per disease show that surveillance workers identified the most suspected cases” makes no sense here, due to your design, indeed the surveillance workers will always detect more suspect cases that the Epi or Response teams (who are coming after a case has been detected by the surveillance worker).

* Please consider the following actions:

* Abstract line 25, methods line 215, Table 3, results lines 240-249: **remove the data quality/validity attribute** as it cannot be evaluated (see above).

* **From lines 241-249, you can move the following slightly rephrased sentence and figures 4 to 7 to the PPV results section**: "Available health facility surveillance data from two MSF health facilities and data from the CBS (suspected cases identified by the surveillance workers and verified by the Epi Alert Team) are shown for AWD, AJS, Diphtheria and measles (Figure 4-7)."

* **CBS case definitions**:

* **CBS workers mostly used the exact same case definitions for suspect cases that at the health facility**, as correctly mentioned lines 182-184: " The same diseases that were included in the CBS were under surveillance at the health facilities using the same MSF case definitions (Table 1)."

* **These were not "community case definitions" as the term is generally understood.** For example, the following ones are community case definitions: acute jaundice: “Any person with a sudden yellowing of the skin for not more than two weeks, with or without elevated body temperature.”; Acute flaccid paralysis: “Any child with a sudden onset of acute paralytic disease." (2014 WHO guide: https://www.afro.who.int/sites/default/files/2017-06/a-guide-for-establishing-community-based-surveillance-102014_0.pdf ). Other examples of guides providing such "community case definitions": Integrated Diseases Surveillance and Response in the African Region. Community-based Surveillance (CBS) Training Manual. WHO, 2015.; Community-based surveillance: guiding principles. IFRC, 2017.; Technical Guidelines for Integrated Disease Surveillance and Response in the African Region; WHO and US CDC, 2010.

* To avoid any misunderstanding, please consider the following actions:

* Line 101: **Replace** "The CBS system used community case definitions (Table 1)" by "The CBS system used standard case definitions (Table 1)".

* Table 1: **Replace column titles**: "CBS case definition (suspected)" by "Case definitions used by CBS surveillance workers" and "Health facility case definition" by "Case definitions used at health facility".

* You may, or not, discuss around lines 347-353 how using "community case definitions" could have improved the ability of community volunteers to detect true disease.

**Minor issues**:

* Abstract line 35 and Discussion line 335:

* "The CBS allowed for timely detection and response [...]". Indeed timely response is correct, but for timely detection this doesn't seem correct: visits were only on a monthly basis and you don't have data on the delay between onset of symptoms and detection.

* Consider rephrasing the sentence to "The CBS allowed for timely response [...]".

* Figure 1:

* You mention in your reply that "We have moved the figure with the overview of the surveillance system to the end of the ‘methods’ section, line 204, as suggested". But, you have not, it is still referred line 73 and its mention doesn't appear later on.

* To clarify that surveillance workers referred to the health facilities and that Epi and Response team reminded visited patients to go to the health facility, consider modifying in the figure: “Referral by Epi Alert Team […]” by “Epi Alert Team or Medical Response Team reminded suspected cases to visit the nearest health facility”.

* “Weekly upload of case information suspected case on EWARS” is unclear, consider “Weekly upload of case information on EWARS” OR “Weekly upload of information on suspected cases on EWARS”.

* Line 58: the wording “Some CBS train community volunteers to use actively search for cases of pre-defined case definition [...]” is unclear, consider the following rewording: "Some CBS train community volunteers to actively search for cases using pre-defined case definitions […]."

* In your reply you mention: "We have added that the households were visited by ‘a surveillance worker’ (single)". But, you have not, only “surveillance workers” (plural) is mentioned in the 4 occurrences of “surveillance workers” visiting households. Consider correcting these four occurrences.

* The number of surveillance workers per team leader is currently mentioned in the results section (line 306), consider moving it to the Methods section, potentially in table 2.

* Figures 4 to 7: you need to add different line types (or use different colours) so that it is easy to understand which line represents what.

* Line 396: don't use the acronym HR, but the full wording: "human resources" (if this is what HR meant).

Reviewer #2: Thank you for your responses. I am satisfied with your responses have met the considerations I had provided and of the other review.

7. PLOS authors have the option to publish the peer review history of their article (what does this mean?). If published, this will include your full peer review and any attached files.

Reviewer #1: **Yes: **José Guerra

Reviewer #2: No

---

## [Author Response · Author response to Decision Letter 1]

2 Dec 2020

Oslo, 30 November 2020

Dear Editor,

We would like to thank you for the consideration of our manuscript ‘Description of Community Based Surveillance in the Rohingya refugee camps in Cox’s Bazar, Bangladesh, 2019’ for publication in PlosOne. In addition, we would like to thank both reviewers very much for taking the time to review our draft manuscript thoroughly and share such insightful comments. We have addressed all comments, kindly find our responses below.

Kind regards, on behalf of the authors,

Elburg van Boetzelaer (corresponding author)

Reviewer #1

Major issues:

• Data quality / validity evaluation attribute:

– In Table 3, the measurement modality for the “Data quality / validity evaluation attribute” is stated as “Comparison of trends of health facility surveillance (gold standards) and community based surveillance trends”.

– The fact is that you can’t compare these trends as they don’t have the same population under surveillance.

* You correctly say it in the discussion lines 357-360: “While the trends of reported suspected cases from all the components of the surveillance system were similar, and could be an indicator of data quality and validity, it must be noted that the catchment population of the CBS and health facility surveillance differed as the health facilities were also visited by

patients who did not live in the CBS catchment area.”

– In any case, it is misleading to evaluate this “Data quality / validity” attribute, as you have no data to do it.

* Lines 240-249: when you compare the “Surveillance worker curve” and

the“Epi alert teamcurve”, you are not evaluating the“data quality/validity” but the evolution of the PPV. The assertion “The trend comparison per diseases how that surveillance workers identified the most suspected cases” makes no sense here, due to your design, indeed the surveillance workers will always detect more suspect cases that the Epi or Response teams (who are coming after a case has been detected by the surveillance worker).

– Please consider the following actions:

* Abstract line 25, methods line 215, Table 3, results lines 240-249: remove the data quality/validity attribute as it cannot be evaluated (see above). We have made the suggested edits.

* From lines 241-249, you can move the following slightly rephrased sentence and figures 4 to 7 to the PPV results section: “Available health facility surveillance data from two MSF health facilities and data from the CBS (suspected cases identified by the surveillance workers and verified by the Epi Alert Team) are shown for AWD, AJS, Diphtheria and measles (Figure 4-7).” We have made the suggested edits.

• CBS case definitions:

– CBS workers mostly used the exact same case definitions for suspect cases that at the health facility, as correctly mentioned lines 182-184: “The same diseases that were included in the CBS were under surveillance at the health facilities using the same MSF case definitions (Table 1).”

– These were not “community case definitions” as the term is generally understood. For example, the following ones are community case definitions: acute jaundice: “Any person with a sudden yellowing of the skin for not more than two weeks, with or without elevated body temperature.”; Acute flaccid paralysis: “Any child with a sudden onset of acute paralytic disease.” (2014

WHOguide: https://www.afro.who.int/sites/default/files/2017-06/a-guide-forestablishing-community-based-surveillance-102014_0.pdf ). Other examples of guides providing such “community case definitions”: Integrated Diseases Surveillance and Response in the African Region. Community-based Surveillance (CBS) Training Manual. WHO, 2015.; Community-based surveillance: guiding principles. IFRC, 2017.; Technical Guidelines for Integrated Disease Surveillance and Response in the African Region; WHO and US CDC, 2010. – To avoid any misunderstanding, please consider the following actions:

* Line 101: Replace “The CBS system used community case definitions (Table 1)” by “The CBS system used standard case definitions (Table 1)”. We have made the suggested edit.

* Table 1: Replace column titles: “CBS case definition (suspected)” by “Case definitions used by CBS surveillance workers” and “Health facility

case definition” by “Case definitions used at health facility”. We have made the suggested edits.

– You may, or not, discuss around lines 347-353 how using “community case

definitions” could have improved the ability of community volunteers to detect true disease. Thank you for this suggestion. However, we have decided not to include this in the discussion as it did not become apparent from this evaluation that the case definitions were an obstacle to community based surveillance workers detecting true disease.

Minor issues:

• Abstract line 35 and Discussion line 335:

– “The CBS allowed for timely detection and response […]”. Indeed timely

response is correct, but for timely detection this doesn’t seem correct: visits

were only on a monthly basis and you don’t have data on the delay between onset of symptoms and detection.

– Consider rephrasing the sentence to “The CBS allowed for timely response […]”. Thank you for pointing this out. We have made the edits.

• Figure 1:

– You mention in your reply that “We have moved the figure with the overview of the surveillance system to the end of the ‘methods’ section, line 204, as suggested”. But, you have not, it is still referred line 73 and its mention doesn’t appear later on. Apologies for this oversight, we have now moved the figure to the end of the methods section.

– To clarify that surveillance workers referred to the health facilities and that Epi and Response team reminded visited patients to go to the health facility, consider modifying in the figure: “Referral by Epi Alert Team […]” by “Epi Alert Team or Medical Response Team reminded suspected cases to visit the

nearest health facility”. We made the suggested edit.

– “Weekly upload of case information suspected case on EWARS” is unclear, consider “Weekly upload of case information on EWARS” OR “Weekly upload

of information on suspected cases on EWARS”. We made the suggested edit.

• Line 58: the wording “Some CBS train community volunteers to use actively search for cases of pre-defined case definition […]” is unclear, consider the following rewording: “Some CBS train community volunteers to actively search for cases using pre-defined case definitions […].” We made the suggested edit.

• In your reply you mention: “We have added that the households were visited by ‘a surveillance worker’ (single)”. But, you have not, only “surveillance workers” (plural) is mentioned in the 4 occurrences of “surveillance workers” visiting households. Consider correcting these four occurrences. We believe we have made the suggested edits (line 106-138).

• The number of surveillance workers per team leader is currently mentioned in the results section (line 306), consider moving it to the Methods section, potentially in table 2. We have added the number of surveillance workers per team leader to Table 2.

• Figures 4 to 7: you need to add different line types (or use different colours) so that it is easy to understand which line represents what. We have made the edits as suggested.

• Line 396: don’t use the acronym HR, but the full wording: “human resources” (if this is what HR meant). We have made the suggested edit.

---

## [Editor Report · Decision Letter 2]

7 Dec 2020

Evaluation of Community Based Surveillance in the Rohinya refugee camps in Cox's Bazar, Bangladesh, 2019

PONE-D-20-14072R2

Dear Dr. van Boetzelaer,

We’re pleased to inform you that your manuscript has been judged scientifically suitable for publication and will be formally accepted for publication once it meets all outstanding technical requirements.

Kind regards,

Jennifer Yourkavitch

Academic Editor

PLOS ONE
---

## [Editor Report · Acceptance letter]

10 Dec 2020

PONE-D-20-14072R2 

Evaluation of Community Based Surveillance in the Rohingya refugee camps in Cox’s Bazar, Bangladesh, 2019 

Dear Dr. van Boetzelaer:

I'm pleased to inform you that your manuscript has been deemed suitable for publication in PLOS ONE. Congratulations! Your manuscript is now with our production department. 

Kind regards, 

on behalf of

Dr. Jennifer Yourkavitch 

Academic Editor

PLOS ONE